# Spatiotemporal Variation in and Responses of the NDVI to Climate in Western Ordos and Eastern Alxa

Hui Zhang [1,†], Jinting Guo [1,2,†] , Xiaotian Li [1], Yajie Liu [1] and Tiejuan Wang [1,2,*]

1   College of Life Sciences and Technology, Inner Mongolia Normal University, Hohhot 010022, China
2   Key Laboratory of Biodiversity Conservation and Sustainable Utilization for College and University of Inner Mongolia Autonomous Region, Hohhot 010022, China
*   Correspondence: wtj105@163.com; Tel./Fax: +86-0471-4392-441
†   These authors contribute equally to this work.

**Abstract:** Vegetation is an important component of the terrestrial ecosystem, and studying the rules of vegetation change and its driving factors is helpful to strengthen the ecological protection and sustainable development of regional vegetation. This study analyzes the changes in Normalized Difference Vegetation Index (NDVI) and its response to climate factors in the five regions of western Ordos and eastern Alxa in China between 2000 and 2020. The MODIS NDVI and meteorological data from 2000 to 2020 was used and the ordinary least squares, trend analysis, and correlation analysis methods were analyzed. The NDVI in this region shows spatial differentiation and is high in the east and low in the west. The overall NDVI has shown a significant increasing trend ($p < 0.01$), and the slope value of the rate of change also shows that the NDVI in 98.17% of the area is increasing. On a temporal scale, NDVI had a significant positive correlation with precipitation ($p < 0.01$), but no significant correlation with temperature changes. On a spatial scale, NDVI was positively correlated with precipitation, which accounted for 95.57% of spatial changes, of which a significant positive correlation accounted for 34.99% ($p < 0.05$). Meanwhile, the temperature and NDVI were negatively correlated but not significantly. A positive correlation accounted for 45.95% of the change, but the insignificant negative correlation accounted for 54.05%. Therefore, comprehensive analysis showed that precipitation played a leading role in the NDVI in the study area. The results are helpful to study the driving mechanism of vegetation growth and provide reference for vegetation protection in regions of western Ordos and eastern Alxa of Inner Mongolia, China.

**Keywords:** climate change; Vegetation; Eastern Alxa; Normalized Difference Vegetation Index; western Ordos

## 1. Introduction

Relevant studies show that the global average temperature rose by 0.72 °C during 1951–2012, and the average temperature of nearly 30 years was the highest in the past 1400 years [1,2]. Vegetation is an important component of the terrestrial ecosystem, and plays an important role in desertification control, soil and water conservation and ecological environment improvement [3,4]. Climate is an important driving factor affecting the dynamic change in terrestrial vegetation. Vegetation is extremely sensitive to climate change, and dynamic changes in vegetation are often used as biological indicators of climate change. Therefore, exploring the spatiotemporal changes of vegetation and its response to climate factors is of great importance for regional ecological protection and sustainable development, as well as for understanding the internal evolution mechanism of terrestrial ecosystems. In the middle and high latitudes of the northern Hemisphere, the extension of the vegetation growing season and the increase in vegetation coverage in recent decades are mainly caused by climate warming. With the deepening of global climate research, vegetation change and its relationship with climate change have become one of the core contents of global change research [5,6].

Remote sensing aims to explore the spatiotemporal dynamics of vegetation and requires the support of long time series data, and remote sensing technology with the advantages of continuous observation and multiple spatiotemporal scales provides a powerful and useful approach to investigating vegetation dynamics. Variations in the Normalized Difference Vegetation Index (NDVI) reflect the global land surface vegetation coverage, which is important for the analysis of the ecological environment. NDVI is calculated based on the principle of strong absorption of green wavelength in the visible spectrum by plant chlorophyll and strong reflection by plant cell tissue in the near-infrared spectrum. Because it is very sensitive to changes in plant biophysical characteristics [7–9], NDVI has the advantages of wide space coverage, high monitoring sensitivity and the partial elimination of atmospheric radiation interference and it has become the best indicator of vegetation growth status and vegetation coverage. NDVI is the most widely used index for characterizing vegetation status [10,11]. It has been applied to the study of vegetation cover change [12,13], driving force analysis [14–16], monitoring, and evaluation [17–20].

Inner Mongolia is one of the most sensitive regions to global climate change [21,22]. Due to the interlaced distribution of different vegetation types, primarily grassland and desert, changes in regional vegetation also have a significant impact on climate change [23]. Although the effects of climatic factors on NDVI have spatial heterogeneity [24,25], relevant research shows that precipitation and temperature are the main factors affecting the changes in NDVI. For example, NDVI fluctuation was found to be closely related to annual precipitation in the Mashhad–Chenaran Plain in northeastern Iran, the Great Plains in central North America, Inner Mongolia in China, and others [26–28]. However, studies in North America and the Yamdrok Tso Basin in Tibet found that air temperature was the dominant factor affecting NDVI [29].

In recent years, many scholars have studied the spatiotemporal variation in NDVI in Inner Mongolia [22,30,31]. A study on the response of NDVI to climate factors also showed that temperature and precipitation were the main factors, and precipitation played a leading role [28,32,33]. However, the above studies [22,28,30–33] were based on the overall span of the Inner Mongolia Autonomous Region and occurred primarily before 2015. Inner Mongolia is narrow and long from east to west, the terrain is complex, and the vegetation distribution varies. The changes in vegetation growth on a spatiotemporal scale vary between different cities and terrains. The present study area, western Ordos and eastern Alxa, is in western Inner Mongolia and spans desert steppes, areas of grassland desertification, and typical desert. There are several national nature reserves in the study area: West Ordos National Nature Reserve, Helan Mountain National Nature Reserve, and Hatengtaohai National Nature Reserve. The West Ordos Nature Reserve is one of eight unique botanical regions in China, containing a variety of rare and endangered species. It is vital to study the response of NDVI to climate changes in the region for biodiversity conservation. And studying the rules of vegetation change and its driving factors is helpful to strengthen the ecological protection and sustainable development of regional vegetation. Thus, the present study examines the average NDVI during the growing season from 2000 to 2020 and uses the ordinary least squares regression, trend analysis, and correlation analysis methods to study the vegetation cover characteristics, growth trends, and the influence of temperature and precipitation on the NDVI. We address the following questions: (1) How did the vegetation cover (NDVI) vary spatiotemporally in western Ordos and eastern Alxa over a 21-year period? (2) Which climate variable (temperature or precipitation) determined the vegetation change? The results of this study provide important information on the relationship between vegetation and climate in western Ordos and eastern Alxa.

## 2. Materials and Methods

### 2.1. Study Area

The study area includes the western Ordos and eastern Alxa region of Inner Mongolia, China. And the study area is located in the mid-latitude region. Geographically, this area extends from 37°25′12″ N to 41°52′12″ N and from 103°22′12″ E to 109°15′ E, which

includes the areas of Alxa Left Banner, Dengkou County, Hangjin Banner, Otog Banner, and Wuhai City (Figure 1). The region has complex topography and diverse landforms. In addition to the undulating sandy and gravel high plain, there are three major deserts, namely Kubuqi Desert, Ulan Buhe Desert and Tengger Desert, as well as dry denudative mountains, such as Wolf Mountain, Table Mountain, Yabhao Mountain, Longshou Mountain and the western slope of Helan Mountain. Gander Mountain and Table Mountain are arranged in a north-south vertical and horizontal manner between the Yellow River and the Ordos Plateau, forming a landform pattern of alternating valleys. The habitat is mainly steppe desert, and the environment is relatively harsh. The northwestern areas are part of the typical desert belt, and the eastern areas are part of the desert steppe belt. The Yellow River crosses the center of the study area (the blue strip between Alxa Left Banner and Wuhai City in Figure 1). The Yellow River irrigation area lies along the river and in Dengkou County, where the vegetation is mainly anthropogenic. It is controlled by the alternate climate of the subtropical high pressure belt and the westerly wind belt, belonging to the typical temperate continental climate, with the characteristics of plateau cold and summer fusion and a large temperature difference between day and night. The average annual temperature of the study area is 16.8–18 °C, which is a warm climate. The annual total precipitation is 100–264 mm, and the water gradient varies between drought and extreme drought.

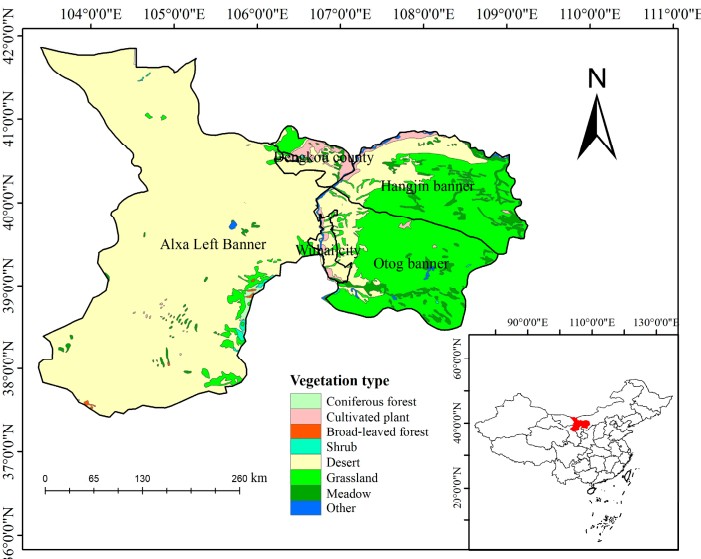

**Figure 1.** Five regions and vegetation types in western Ordos and eastern Alxa.

## 2.2. Data Sources

### 2.2.1. NDVI

NDVI data were derived from the monthly maximum synthetic Vegetation Index Product (MOD13Q1) data developed by the MODIS Land Product Group of the U.S. Geological Survey, based on a unified algorithm. The dataset involved atmospheric calibration and geometric corrections, especially errors produced by satellite alternation, to ensure data quality. MOD13Q1 monthly dataset composites were generated using the maximum value composite (MVC) approach to further reduce the influence of clouds, atmosphere, and solar zenith angle. The MVC method selects the largest NDVI per pixel. The compound algorithm reduces the effect of angular and sun-target-sensor variations and provides robust spectral measures of the amount of ground vegetation, allowing precise comparison of spatiotemporal variations in terrestrial photosynthetic activity. The spatial resolution is 250 m, and the time period is 2000–2020. To avoid the abnormal NDVI caused by winter snow cover, this study selected the growing season data from April–October, which has sufficient vegetation growth for research [34]. Additionally, pixels with low credibility were

excluded. We calculated the mean growing-season NDVI by averaging monthly maximum values from April through October. Pixels at which the average growing season NDVI was <0.05 were masked as nonvegetated areas.

### 2.2.2. Meteorological Data

Meteorological data came from the China Meteorological Administration Data Center, http://data.cma.cn/ (accessed on 3 May 2021). The average temperature and total precipitation from April to October in 21 years (2000–2020) of 34 stations in the study area and nearby were selected. The monitoring data of meteorological stations around the study area were interpolated by Kriging interpolation using KQGIS V8.1 software [35]. Finally, raster data consistent with NDVI spatial resolution and projection were obtained.

### 2.3. Research Methods

### 2.3.1. Analysis of Vegetation Coverage Characteristics

The average NDVI data during the growing season from 2000 to 2020 were used to study the dynamic changes in vegetation cover in the study region, and the spatial distribution characteristics of the NDVI were analyzed combined with a vegetation type map. The interannual trends in NDVI and climate factors throughout the study area were calculated using the ordinary least squares regression analysis method [36].

$$y = at + b + \varepsilon \tag{1}$$

$$a = \frac{\sum\limits_{i=1}^{21} (y_i - \bar{y})(t_i - \bar{t})}{\sum\limits_{i=1}^{21} (y_i - \bar{y})^2} \tag{2}$$

In the formulas, $y$ is the NDVI or climatic factor value, $a$ represents the interannual variation trend in the NDVI or climatic factors throughout the region, $t$ is time, $b$ is intercept, $\varepsilon$ is random error, $\bar{y}$ is the average y, $\bar{t}$ is the average time.

### 2.3.2. Trend Analysis Method

The change in NDVI in the study area during the 21 years was analyzed at the pixel scale using the trend analysis method [37]. The formula is as follows:

$$Slope = \frac{n \times \sum\limits_{i=1}^{n} i \times NDVI_i - \sum\limits_{i=1}^{n} i \sum\limits_{i=1}^{n} NDVI_i}{n \times \sum\limits_{i=1}^{n} i^2 - \left(\sum\limits_{i=1}^{n} i\right)^2} \tag{3}$$

where $n$ is the length of the time series, $i$ represents the year in the study period, $NDVI_i$ is the NDVI value of the $i$th year, and *Slope* is the vegetation change trend for the pixel. When *Slope* > 0, the NDVI for that pixel increased during the study period. Similarly, when *Slope* < 0, the NDVI decreased during the study period, and when *Slope* = 0, the NDVI was unchanged.

### 2.3.3. Pearson Correlation Analysis

The correlations between the NDVI, temperature, and precipitation in the study area from 2000 to 2020 were analyzed using the Pearson correlation coefficient (*r*) [37]. Correlations ranged from +1 to −1. Zero correlation indicated no relationship between the variables. A negative correlation indicated that as one variable increased, the other

declined. A positive correlation indicated that both variables changed in the same direction. The formula is:

$$r_{xy} = \frac{\sum\limits_{i=1}^{n} \left[ (x_i - \overline{x})(y_i - \overline{y}) \right]}{\sqrt{\sum\limits_{i=1}^{n} (x_i - \overline{x})^2 \sum\limits_{i=1}^{n} (y_i - \overline{y})^2}} \tag{4}$$

The $n$ is the length of the time series, $i$ is the year in the study period, $x_i$ is the NDVI value of the $i$th year, $y_i$ is the temperature or precipitation value of the $i$th year, $\overline{x}$ is the mean NDVI for 2000–2020, and $\bar{y}$ is the mean precipitation or mean temperature for 2000–2020.

## 3. Results

### 3.1. Spatial Distribution Characteristics of the NDVI

From 2000 to 2020, the annual average NDVI in the study area was high in the east and low in the west (Figure 2). Because the study area was arid, the overall NDVI was low. The proportion the study area with NDVI < 0.2 reached 91%. Areas with NDVI > 0.1 accounted for 59.44% and were mainly distributed in the eastern and southern areas. Areas between 0.1 and 0.2 accounted for 50.44% of the total area and were mainly distributed in the central, southern, and eastern parts of the study area. The vegetation types there were mainly desert, steppe, and meadow vegetation. The areas with NDVI > 0.2 accounted for 9.00% of the study area and were mainly distributed in the north-central region (agricultural irrigation area in Dengkou County), a narrow and long area in the middle of the study area (along the Yellow River), the south-central study area (southeast of Alxa Left Banner, mainly Helan Mountain), and the eastern areas (Hangjin Banner and eastern Otog Banner). The vegetation types in these areas were mountain coniferous forest, mountain broadleaf forest, mountain shrub, meadow, anthropogenic vegetation, and desert steppe (Figure 1, Figure 2). NDVI values below 0.1 accounted for 40.56% of the study area (Figure 2) and were mainly distributed in western Alxa Left Banner, which is an arid desert area where the vegetation coverage is very low. Thus, overall NDVI shows noticeable spatial differentiation.

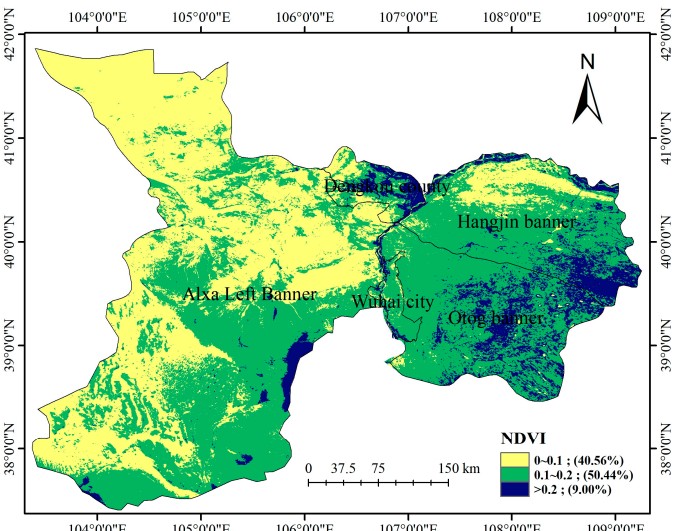

**Figure 2.** Spatial distribution of the average NDVI during the growing season in the five regions of western Ordos and eastern Alxa from 2000 to 2020.

### 3.2. Spatial and Temporal Variation in the NDVI during the Growing Season

As shown in Figure 3, the average NDVI showed a trend of fluctuation and significant increase ($p < 0.01$) during the study period. The average NDVI value increased from 0.1044 in 2000 to 0.1402 in 2020, with an average annual increase of 0.0017 and an annual growth rate of 1.70%. The vegetation cover in the study area increased as a whole. Spatial dynamics

of the vegetation cover are depicted in Figure 4a. There was strong spatial heterogeneity from the per-pixel analysis. Pixels with increasing trends (*Slope* > 0) accounted for 98.17% of the study area, which were mostly found across the study area, with the exception of the scatter areas of Alxa Left Banner and Wuhai city (Figure 4a). Pixels in 85.45% of the study area showed a significant increase ($p < 0.05$) (Figure 4b). The NDVI in 1.83% of the study area decreased (Figure 4a), mainly in scattered areas in northern Alxa Left Banner and northwestern Wuhai City. This was mainly related to human activities, in particular, the construction of the Yellow River water conservancy project, which formed Wuhai Lake in 2013, thereby decreasing the vegetation coverage.

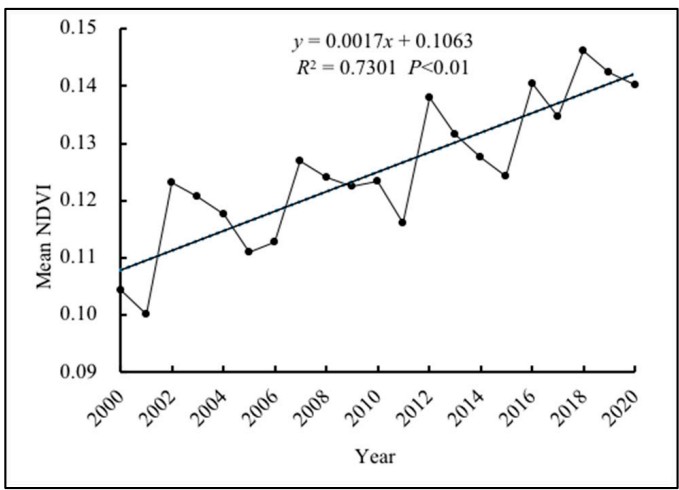

**Figure 3.** Annual variation in the average NDVI during the growing season in the five regions of western Ordos and eastern Alxa from 2000 to 2020.

### 3.3. NDVI and Driving Force Analysis of Climate Factors

3.3.1. Correlation between Growing-Season Mean NDVI and Climate Variables at Scale of Entire Study Area

During 2000–2020, the highest average temperature during growing season was 18.01 °C (2017), the lowest was 16.80 °C (2003), and the average temperature during growing season was 17.40 °C (Figure 5). The highest total precipitation was 264.19 mm (2012), the lowest was 100.41 mm (2005), and the average total precipitation during growing season was 184.18 mm (Figure 5). Precipitation showed an increasing trend, but temperature remained steady, only changing slightly (16.8–18.0 °C). From the overall trend for the 21 years, the change in the NDVI was basically consistent with that of the precipitation (Figure 5). At the entire study area scale, we analyzed the correlation between the NDVI and climate variables (Table 1). The NDVI was positively and significantly correlated with total precipitation of growing season ($R = 0.582$; $p < 0.01$), but weakly and positively related with mean temperature of growing season ($R = 0.142$; $p > 0.05$). We compared correlation coefficients between NDVI and climate variables, finding that correlation between NDVI and precipitation was much stronger than between NDVI and temperature. Thus, we conclude that precipitation was the primary determinant of vegetation dynamics in the western Ordos and eastern Alxa regions. In addition, Figure 5 shows that the precipitation was highest in 2012, 2018, and 2002; the NDVI also showed a small peak in these years. Although it was not relatively high in 2002, it continued to rise and was highest in 2018. Thus, the precipitation increased gradually over the study period and had a lasting impact on the vegetation coverage in the study area.

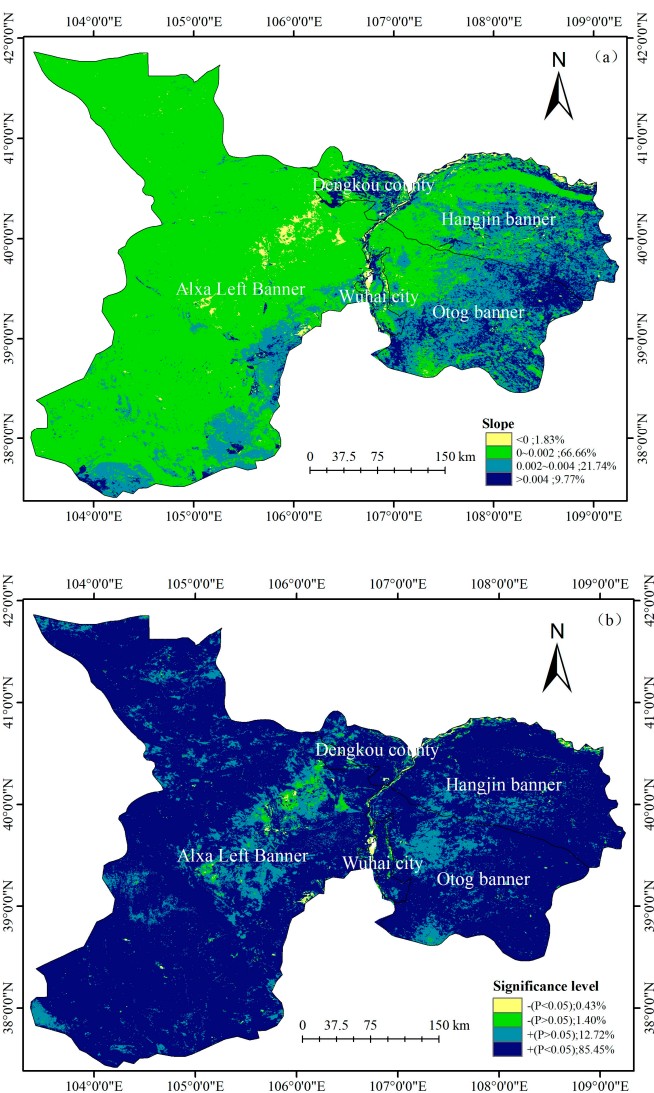

**Figure 4.** Spatial variation trends in average NDVI: (**a**) magnitude and (**b**) statistical test result at 5% significance level during the growing season in the five regions of western Ordos and eastern Alxa from 2000 to 2020.

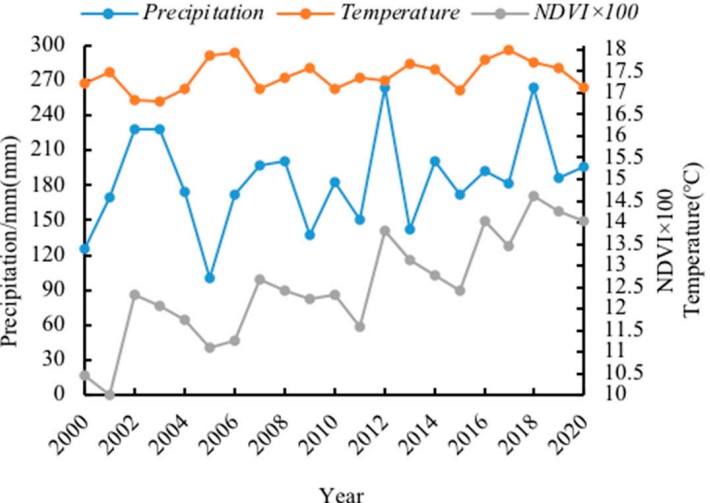

**Figure 5.** Annual variation in the NDVI, temperature, and precipitation in the five regions of western Ordos and eastern Alxa from 2000 to 2020.

**Table 1.** Correlation of NDVI with precipitation and temperature on the scale of entire study area.

|      | Precipitation | Temperature |
|------|---------------|-------------|
| NDVI | 0.582 **      | 0.142       |

Note: ** indicates extremely significant correlation ($p < 0.01$).

3.3.2. Correlation between Growing-Season Mean NDVI and Climate Factors at Pixel Scale

To further assess correlations between NDVI and climate variables, we calculated correlation coefficients between NDVI and the two climate factors at all pixels (Figure 6). The spatial relationship between NDVI and air temperature during the study period is shown in Figure 6a. Using a significance level of $p < 0.05$, the results showed that the correlation between air temperature and NDVI was not significant, among which 45.95% of the NDVI showed an insignificant negative correlation and 54.05% showed an insignificant positive correlation. As shown in Figure 6b, 95.57% of the NDVI was positively correlated with the spatial variation in precipitation, among which 34.99% was significant ($p < 0.05$) and 18.12% was extremely significant ($p < 0.01$). This was mainly distributed in most parts of Otog Banner, the southern part of Hangjin Banner, the southernmost part of Alxa Left Banner, and parts of the northern study area. Only 0.05% of the NDVI showed a significant negative correlation with precipitation. Therefore, that precipitation was the dominant factor in the growth of vegetation in the study area.

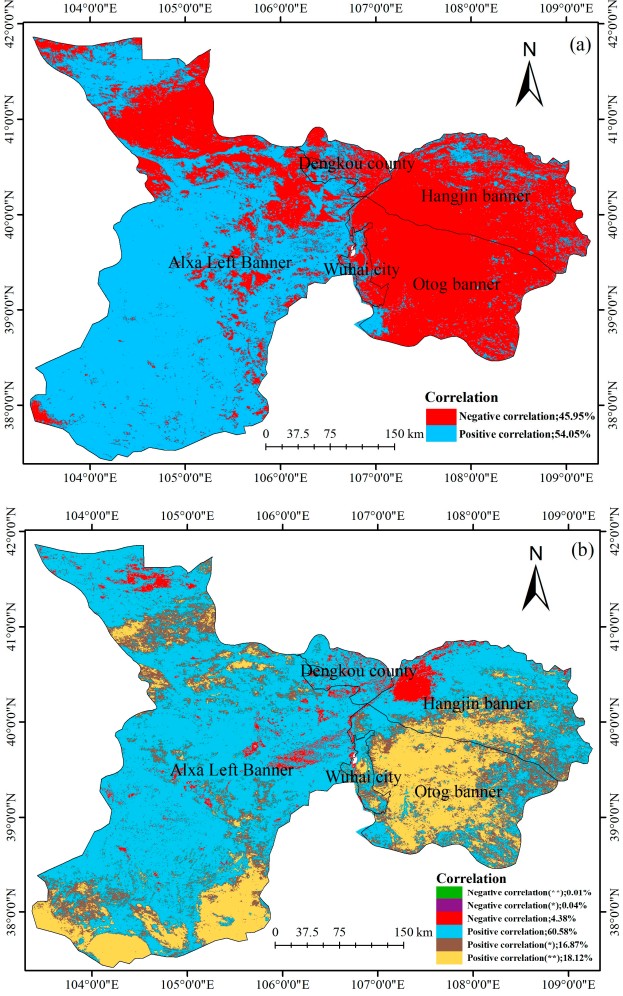

**Figure 6.** Spatial distribution of the correlation between (**a**) NDVI and average temperature, and (**b**) NDVI and total precipitation during the growing season in the five regions of western Ordos and eastern Alxa from 2000 to 2020. * indicates $p < 0.05$, and ** indicates $p < 0.01$.

## 4. Discussion

From the spatial distribution, the overall NDVI in the five regions of western Ordos and eastern Alxa was low, which consistent with the result of Guo [38]. The proportion of NDVI < 0.2 was 91%, and the proportion of NDVI < 0.1 was 40.56%. These low values are due to the vegetation types in the study area, which are mainly desertified grasslands [39]. The Alxa Left Banner, which accounts for more than half of the study area, contains desert areas, so the overall NDVI value in the study area was low. However, from a temporal perspective, the NDVI in the study area showed a significant increasing trend from 2000 to 2020 ($p < 0.01$), and the vegetation coverage in the study area gradually increased, which was closely related to climate change and the national emphasis on ecosystem protection [40,41].

Temperature and precipitation are the main factors affecting the distribution of vegetation on land. Related studies also show that changes in NDVI are related to these factors [42–44]. Vegetation in the cold regions of high latitudes is mainly affected by temperature; for example, the NDVI in Northeast China increases significantly with the increasing average surface temperature during the growing season [45]. Xue et al. also suggested that the warming climate resulted in significant changes in land cover types and plant biomass in the northern high latitudes [46]. However, vegetation in arid areas is mainly affected by precipitation, such as the positive correlation between precipitation and the NDVI in the arid, mountainous areas of Armenia [47]. The western region of Inner Mongolia, in which the study area is located, is also an arid area. The increasing vegetation cover in the study area was caused by increasing precipitation, which increase effective moisture for vegetation growth [36]. Su et al indicated that the extreme precipitation index had a greater impact on the vegetation growth and changes than the extreme temperature index in the Inner Mongolia [48]. Erdengerel et al. studied the vegetation cover of the West Ordos National Nature Reserve in the study area from 2001 to 2013 [49]. Their results showed that the correlation coefficient between precipitation and NDVI was 0.66, and it was a significant correlation ($p < 0.01$). The present study results are highly consistent with the former study (the correlation coefficient was 0.582, $p < 0.01$), which shows that the increased vegetation cover was related to the increased precipitation. Furthermore, in recent years, the state has strengthened the protection of nature reserves, ecological function areas, grasslands, woodlands, wetlands, and wildlife, and strengthened the comprehensive management of deserts, sandy land, and soil erosion. There are currently three national nature reserves in this study area, and thus human protection has also played an effective role in vegetation restoration for many years.

## 5. Conclusions

Based on the NDVI, precipitation, and temperature data in the five regions of western Ordos and eastern Alxa from 2000 to 2020, the response of NDVI to meteorological factors was analyzed on temporal and spatial scales, pixel by pixel. The main conclusions are as follows:

There is spatial differentiation in the NDVI in the area, decreasing from southeast to northwest. NDVI < 0.1 occurred in 40.56% of the study area, mainly in the western Alxa Left Banner. NDVI > 0.1 occurred in 59.44% of the study area, mainly in the eastern and southern areas.

Over the study period, the NDVI significantly increased ($p < 0.01$), with an average annual increase of 0.0017 and an annual growth rate of 1.70%. Additionally, the overall vegetation coverage in the study area increased. On a spatial scale, the NDVI of 98.17% of the study area showed an increasing trend, of which 85.45% was significant ($p < 0.05$), and 1.83% showed a decreasing trend, mainly distributed in Alxa Left Banner and Wuhai City.

During the 21 years, the precipitation in the study area showed an increasing trend, which was consistent with the trend in the NDVI; the two showed a very significant positive correlation ($p < 0.01$). Spatially, the NDVI was positively correlated with precipitation changes, accounting for 95.57% of the change, of which 34.99% had a significant positive

correlation ($p < 0.05$), and 18.12% had a very significant positive correlation ($p < 0.01$). There was no significant change in temperature, and there was no significant correlation between temperature and NDVI. Therefore, precipitation was the main control factor affecting vegetation coverage in the study area; an increase in precipitation promoted an increase in vegetation coverage.

**Author Contributions:** Software, J.G.; Formal analysis, X.L. and Y.L.; Data curation, H.Z., J.G., X.L. and Y.L.; Writing—original draft, H.Z.; Writing—review & editing, H.Z., J.G. and T.W.; Supervision, J.G. and T.W.; Funding acquisition, T.W. All authors have read and agreed to the published version of the manuscript.

**Funding:** Science and Technology Project of Inner Mongolia Autonomous Region (2020GG0124), Natural Science Foundation of Inner Mongolia Autonomous Region (2020BS03043) and Cooperative Education Project of Ministry of Education (Construction of University-Enterprise Joint Laboratory Based on Localization KQGIS Platform).

**Institutional Review Board Statement:** Not applicable.

**Informed Consent Statement:** Not applicable.

**Data Availability Statement:** Not applicable.

**Acknowledgments:** The author would like to thank the Science and Technology Project of Inner Mongolia Autonomous Region (2020GG0124), China for supporting my paper. Thanks to remote sensing Data source MODIS Land Product Group of the U.S. Geological Survey and Chinese Meteorological Administration Data Center. Hui Zhang and Jinting Guo contributed to the work equally and should be regarded as co-first authors.

**Conflicts of Interest:** The authors declare no conflict of interest.

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
