# Peer review of "Spatiotemporal Variation in and Responses of the NDVI to Climate in Western Ordos and Eastern Alxa"

_sustainability, doi:10.3390/su15054375_

Round 1

Reviewer 1 Report

This paper analyzes the changes in NDVI and its response to climate factors in the study area from 2000 to 2020. The method used in this paper is practical and the results obtained are reasonable. But there is a question, for readers of sustainability, what research implications can this article bring to the rest of the world? Here are some more comment details:

1.      It is necessary to introduce the terrain of the study area.

2.      Meteorological data is the same as the remote sensing data from page 3 line 90-95, so I don't know how the author uses the data. And this part is very important.

3.      Since NDVI is related to precipitation and temperature, it is recommended to analyze the monthly variation and correlation during the growth season.

4.      It is suggested that the author analyze and apply the research results to practice.

Author Response

Revision notes:

Dear Reviewer:
We would like to thank you for all the valuable and constructive comments, which were very helpful to the improvement of our manuscript. The manuscript has been revised accordingly based on the comments and suggestions. Our point-to-point responses to your comments are listed as follows, with the original comments in italic and our responses in blue. The revised manuscript is also attached with trackable changes as well as a clean version. 
We believe that our revised manuscript will benefit the readership of Sustainability and look forward to hearing from you soon.
Sincerely yours,
Jinting Guo

Response to Comments of Reviewer

It is necessary to introduce the terrain of the study area.

Response: We thank Reviewer for the valuable comments. We have taken these advices and revised them based on the suggestions. And we added the terrain of the study area in the revised manuscript (Page 2 lines 71-78).

Meteorological data is the same as the remote sensing data from page 3 line 90-95, so I don't know how the author uses the data. And this part is very important.

Response: We thank Reviewer for the constructive comments. We well-acknowledge this point and have fully-incorporated the suggestions into our revised manuscript. And we added the description of meteorological data in part of Materials and Methods in the revised manuscript (Page 3 lines 96-102).

Since NDVI is related to precipitation and temperature, it is recommended to analyze the monthly variation and correlation during the growth season.

Response: We thank Reviewer for the valuable comments. NDVI is related to precipitation and temperature and we analyzed the relationship between the NDVI and average temperature and total precipitation during the growing season. Some former researches employed that the NDVI in the growing season is most closely related to the average temperature or total precipitation during the growing season (Mao et al., 2012; Guo et al., 2017).

References:

MAO D.H., WANG Z.M., LUO L., REN C.Y. Integrating AVHRR and MODIS data to monitor NDVI changes and their relationships with climatic parameters in northeast china. Int. J. Appl. Earth Obs. 18, 528, 2012.

GUO J.T., HU Y.M., XIONG Z.P., YAN X.L., REN B.H., BU R.C. Spatiotemporal Variations of Growing-season NDVI Associated with Climate Change in Permafrost Zone of Northeast China. Pol. J. Environ. Stud. 26(4), 1521-1529, 2017.

It is suggested that the author analyze and apply the research results to practice.

Response: We thank Reviewer for the constructive comments. We have taken this advice and added the description and analysis of our research results. Moreover we added the relevance of our analysis results to practice in the revised manuscript (Page 5 lines 161-165; Page 6 lines 181-196; Page 7 lines 212-214; Page 8 lines 231-233; Page 9 lines 251-255).

Reviewer 2 Report

* Introduction: very good section, however, authors should need to update this section with more recent examples and I suggest developing the Introduction chap. also based on some projects results/official reports.
* The research questions should be more developed.

* What does it add to the subject area compared with other published material?

* Make sure to discuss your Conclusions in relation to other international studies hypotheses.

*The connection between results and some research projects could also be approached this could lead to an interesting and policy-relevant discussion. The discussion section should be improved by citing further references from international researchers.

 Overall, the manuscript displays an original work.

Author Response

Revision notes:

Dear Reviewer:
We would like to thank you for all the valuable and constructive comments, which were very helpful to the improvement of our manuscript. The manuscript has been revised accordingly based on the comments and suggestions. Our point-to-point responses to your comments are listed as follows, with the original comments in italic and our responses in blue. The revised manuscript is also attached with trackable changes as well as a clean version. 
We believe that our revised manuscript will benefit the readership of Sustainability and look forward to hearing from you soon.
Sincerely yours,
Jinting Guo

Response to Comments of Reviewer

Introduction: very good section, however, authors should need to update this section with more recent examples and I suggest developing the Introduction chap. also based on some projects results/official reports.

Response: We thank Reviewer for the valuable comments. We have taken this advice and added some examples based on recent results (Page 1 lines 29-33).

The research questions should be more developed.

Response: We thank Reviewer for the valuable comments. We had made more development of the research questions in the revised manuscript (Page 2 lines 68-70).

What does it add to the subject area compared with other published material?

Response: We thank Reviewer for the valuable comments. Changes in terrestrial vegetation due to the impact of climate change and environment directly affect vegetation NDVI, and changes in vegetation NDVI can in turn characterize changes in terrestrial vegetation. Our study area is an ecologically fragile area, and the vegetation changes in this area are rarely studied. In the face of a deteriorating environment, we can precisely learn about surface vegetation coverage and reveal the pattern of surface spatial change by evaluating the influence of climate change and other factors on the process of vegetation growth. This is significant for identifying the causes of vegetation change and analyzing the regional ecological environment.

Make sure to discuss your Conclusions in relation to other international studies hypotheses.

Response: We thank Reviewer for the valuable comments. We have taken this advice and added some discussion based on other international studies hypotheses in the revised manuscript (Page 5 lines 161-165; Page 6 lines 181-196; Page 7 lines 212-214; Page 8 lines 231-233; Page 9 lines 251-255).

The connection between results and some research projects could also be approached this could lead to an interesting and policy-relevant discussion. The discussion section should be improved by citing further references from international researchers.

Response: We thank Reviewer for the valuable comments. We have taken this advice and added some discussion and references based on other international studies hypotheses in the revised manuscript (Page 8 lines 231-233; Page 9 lines 251-255).

Round 2

Reviewer 2 Report

The manuscript is acceptable in the present form.